# VIPHY: Probing "Visible" Physical Commonsense Knowledge

**Shikhar Singh    Ehsan Qasemi    Muhao Chen**
University of Southern California
ssingh43@usc.edu

## Abstract

Vision-language models (VLMs) have shown remarkable performance on visual reasoning tasks (e.g. attributes, location). While such tasks measure the requisite knowledge to ground and reason over a given visual instance, they do not, however, measure the ability of VLMs to retain and generalize such knowledge. In this work, we evaluate VLMs' ability to acquire "visible" physical knowledge – the information that is easily accessible from images of static scenes, particularly along the dimensions of object color, size, and space. We build an automatic pipeline to derive a comprehensive knowledge resource for calibrating and probing these models. Our results indicate a severe gap between model and human performance across all three dimensions. Furthermore, we demonstrate that an LM tuned on the captions significantly outperforms VLMs on both size and spatial tasks – highlighting that despite sufficient access to ground language with visual modality, they struggle to retain such knowledge. The dataset and code are available at https://github.com/luka-group/ViPhy.

## 1 Introduction

The ability to reason and acquire knowledge from experience, while being intuitive for humans, has been a long-standing challenge for AI agents (McCarthy et al., 1960). Examples such as the color of grass, or the relative position of monitor and table, are formally regarded as commonsense knowledge (Chi, 2005). The retention of such knowledge in humans is achievable due to the presence of long-term memory, broadly classified into *episodic* and *semantic* memory (Tulving, 1972; Camina and Güell, 2017). While the former stores the information pertaining to personal events, the latter is geared towards general, decontextualized knowledge.[1] Prior studies (Greenberg and Verfaellie,

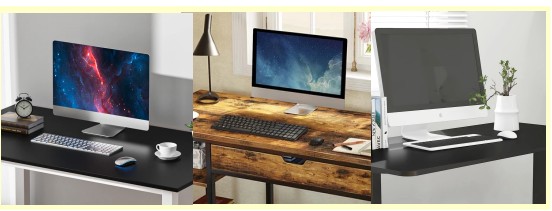
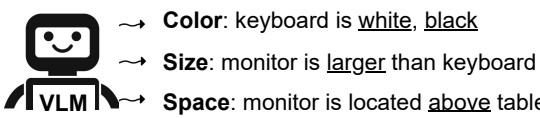

→ **Color**: keyboard is white, black
→ **Size**: monitor is larger than keyboard
→ **Space**: monitor is located above table

Figure 1: We propose VIPHY for probing the ability to generalize visually accessible knowledge – particularly along the dimensions of color, size, and space.

2010) have acknowledged the interdependency between them, particularly the *consolidation* of semantic knowledge from episodic memories – aids humans to acquire commonsense from experience.

Pretrained language models (Devlin et al., 2019; Raffel et al., 2020) have demonstrated the capacity to reason (Wang et al., 2019) and retain knowledge (Petroni et al., 2019; Da et al., 2021). Likewise, vision-language models (Lu et al., 2019; Radford et al., 2021) driven by the availability of large-scale paired image-text datasets have shown strong performance on visual reasoning tasks (Antol et al., 2015; Chen et al., 2015). While such tasks emphasize the model's ability to draw inferences from a specific visual instance – primarily to ground entities and reason about their attributes and relations, they do not, however, explicitly measure the consolidation of such knowledge.[2] In this work, we evaluate the model's ability to generalize aspects of grounding and reasoning tasks, regarded as commonsense knowledge.

Prior works have been largely directed towards probing language models pertaining to object properties such as weight, size, speed, and affor-

---

[1] For instance, the memory of one's birthday cake is episodic, whereas knowing that most birthdays include a cake is part of semantic memory.

[2] Counting bike wheels for different instances is an example of reasoning while generalizing that bikes have two wheels is consolidating across instances.

dance (Forbes and Choi, 2017; Forbes et al., 2019). Drawing upon the notion of world scopes (Bisk et al., 2020a), we find that such datasets, albeit comprehensive across aspects of physical knowledge, are ideally suited for embodied agents capable of interacting with the physical environment. This motivates us to develop resources that better align with the world scope of existing AI systems, primarily vision-language models.

In this work, we introduce VIPHY, a **vi**sible **phy**sical commonsense dataset designed to probe aspects of physical knowledge that are easily accessible in images of static scenes. Therefore, it can be argued that models pre-trained on such data have sufficient access to the "visible world". We build a large-scale dataset along three dimensions of objects: (1) color, (2) size, and (3) space. In contrast to prior works (Paik et al., 2021; Zhang et al., 2022), we bypass crowdsourced annotations in favor of an automated pipeline to derive a resource spanning 14k objects (30×) from raw images. This is achieved by extracting object subtypes – informed by the visual context in images (e.g. kitchen sink). We leverage image data, along with existing vision-language and depth perception models to develop VIPHY.

Beyond scale, we introduce a resource for probing spatial knowledge of common environments. Although one can reason along several types of spatial relations for a visual instance (e.g. a cat *behind* a laptop; Liu et al. (2022a)), we find that mapping them to commonsense knowledge is non-trivial.[3] We define spatial relations by selecting "ground" as the observer and specifying the relative elevation of objects under an allocentric reference frame (Klatzky, 1998).

We probe state-of-the-art models on VIPHY, and find a significant gap across all three dimensions, compared to human performance. Previous works (Paik et al., 2021; Liu et al., 2022b) have corroborated the improvements from language grounding towards acquiring visual knowledge – our results, however, show a more nuanced picture. While VLMs fare much better than LMs on recalling colors, the caption pretrained baseline (Zhang et al., 2022) significantly outperforms VLMs on both size and spatial inference tasks. This highlights that despite access to visual modality, existing VLMs struggle to effectively consolidate such

---

[3]Due to challenges in specifying the reference frame of the observer, the canonical pose of the objects, and the situational nature of a scene.

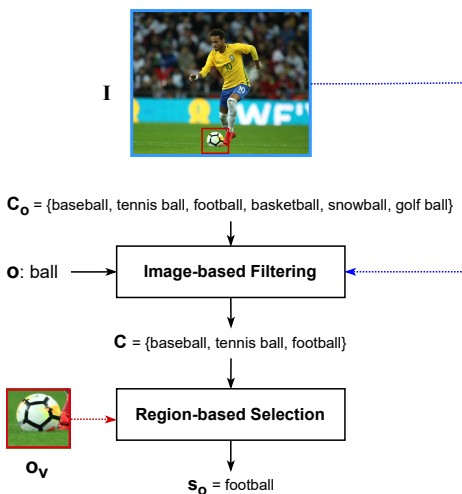

Figure 2: Subtype Selection Module: Given object $o$ in image $I$, assigns subtype $s_o$ from candidate set $C_o$.

knowledge.

The contributions of this work can be summarized as follows: (1) We build a comprehensive dataset, covering multiple aspects of visually accessible knowledge (§3), which is developed through an automated pipeline to derive high-quality resources from images at scale (§2). (2) We conduct extensive benchmarking across several state-of-the-art language and vision-language models and find significant gaps in human performance (§4). (3) We demonstrate a baseline tuned on the caption that significantly outperforms its vision-language counterparts – highlighting that despite access to images, VLMs struggle to consolidate such knowledge.

## 2 Pipeline

We provide a conceptual overview of our pipeline for developing VIPHY, as illustrated in Fig. 3. During the preprocessing stage, we build an internal database comprising object names and corresponding subtype candidates. Given image and object regions as input[4], we substitute object names with their subtypes (§2.1), and compute the corresponding depth map. The processed data is used to extract color, size and spatial knowledge (§2.2).

### 2.1 Object Subtype

While object recognition datasets consider a wide range of objects, such tasks do not necessitate fine-grained categories (Zou et al., 2019). However,

---

[4]Available from either manual annotations or object detection/segmentation model (e.g. He et al. (2017)).

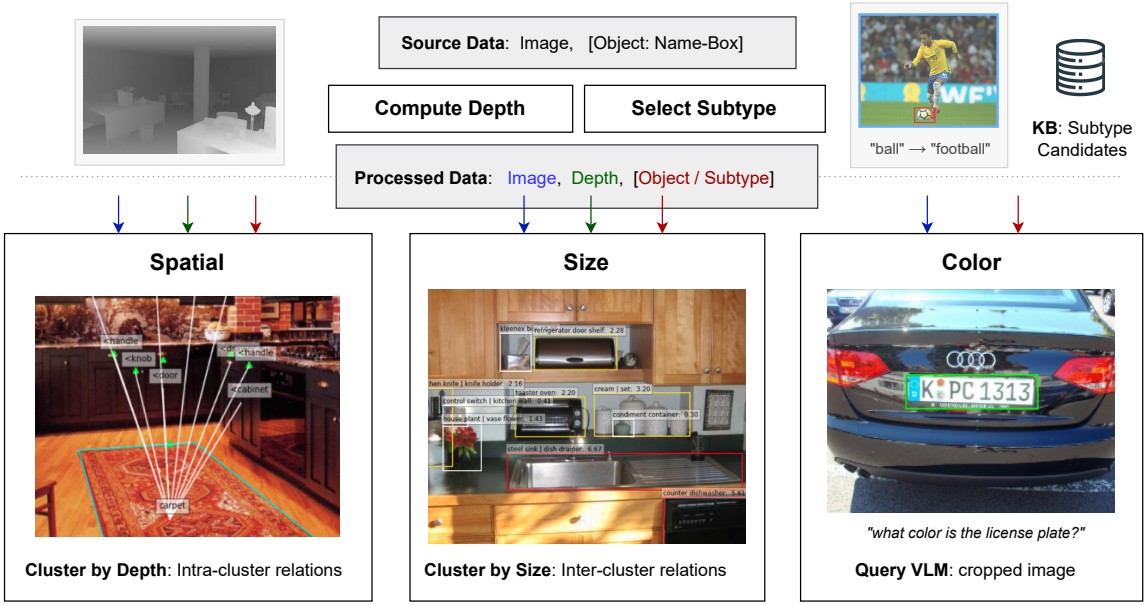

Figure 3: Pipeline Overview: The preprocessing step computes the image depth map, and re-annotates objects by selecting the best subtype from the set of candidates (KB). The color, size and spatial knowledge are then derived independently.

object *subtypes* inform attributes such as color, and help contextualize objects in absence of visual signals (e.g. office chair). Although subtypes are generally accessible from knowledge bases (KB), their coverage is often limited.[5] We extend this definition to include objects defined by visual context – indicating event, location, state, part, etc. (Appendix Tab. 9). For **subtype collection**, we parse captions to build a set of object names. We then employ suffix-based lexical matching to derive subtypes for each object, and merge with hyponyms from knowledge base. The resulting data represents a mapping between the object name and its candidate subtypes.

As our goal is to derive object attributes and relations directly from images, we design a **subtype selection** module to annotate the source image regions with the best subtype. This is required since human annotators often abstract the object name to avoid redundancy when presented with visual context (example in Appendix Fig. 12) – congruent with the maxim of quantity (Grice, 1975). Likewise, existing object detectors are not suited for open-vocabulary and fine-grained classification (Minderer et al., 2022).

The module is designed to query from subtype candidates using visual features. It employs a two-stage approach to filter candidates using image context, and select the best subtype with region-level features, as illustrated in Fig. 2. The visual and textual inputs are embedded using a dual stream vision-language model. Formally, given the visual feature of the image $I$, textual features of the object $o$ and subtype candidates $C_o$, we extract the appropriate subtype as follows:

$$C = \{c | c \in C_o, sim(c, I) > sim(o, I)\} \cup o$$

Here, $sim(.)$ is the cosine similarity. Intuitively, since the object name is independent of visual context, it serves as an anchor for excluding subtypes that do not align with the contextual cues. In the next stage, we incorporate visual features of the object region $o_v$, to query from filtered candidate set $C$, and compute the best subtype $s_o$:

$$s_o = \arg\max_{c \in C} sim(o_v, c)$$

The preprocessed dataset comprises object-subtype mapping for every bounding box region in the image.

## 2.2 Knowledge Extraction

Given the image and depth map, along with object-subtype region annotations, we independently extract color, size and spatial knowledge.

---

[5]We report ~60% object name overlap between ConceptNet KB (Speer et al., 2017) and our collection – derived from dense captions in Visual Genome (Krishna et al., 2017).

**Color** Prior works ([Paik et al., 2021](#)) have relied on human annotations to acquire the color distribution of objects instead of inferring color from pixel values due to challenges such as lighting, shadow, segmentation, etc. However, we argue that large-scale availability of images can mitigate potential noise associated with automated extraction. Given the ubiquity of color attribute in visual reasoning tasks ([Antol et al., 2015](#); [Hudson and Manning, 2019](#)), we find that VLMs pretrained on such datasets are reliable for inferring color from images. As object localization is decoupled from attribute recognition in the pipeline, the input to the VLM is simply the cropped image region, queried with a predefined textual prompt (detailed in §3.1).

**Size** To derive size relations, we consider co-occurring objects in a scene. As objects in an image are expected to appear at varying depths, we approximate perceived size by including scene depth. Given an image, depth map and object-region annotations as inputs, the objects are clustered by size – defined as the bounding box area scaled by mean depth of the region. The sorted partitions are then used to derive inter-cluster relations. The object pair relations are aggregated across images. The number of clusters are fixed for all instances.

**Spatial** We define spatial knowledge as the relative elevation between objects, for a given scene type. To infer these relations directly from image, however, is challenging as perspective projection of 3D world distorts the relative elevation due to variation in depth. We discount this distortion by partitioning the image by depth, and compute *intra-cluster* object relations, *i.e.* we discard the depth coordinate of objects that belong to the same cluster, and simply compare the relative elevation. The *inter-cluster* relations are derived transitively via overlapping partitions – defined by objects with dual membership, as illustrated in [Fig. 4](#). The spatial relations are aggregated across all images for a given scene type. We detail the specifics of mapping object annotations to spatial relations in Appendix [A.3](#).

## 3 Dataset

This section details the specific data sources and models used to develop VIPHY (§3.1). We also report the dataset statistics and task format for each dimension (§3.2). Additional parameters related to dataset construction are provided in Appendix [A.1](#).

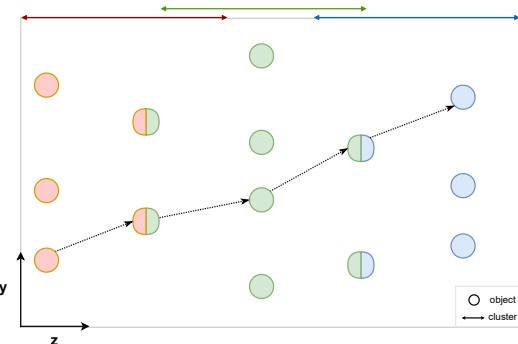

Figure 4: Illustrates transitive spatial relation, computed across partitions (ordered by depth). The y-axis denotes elevation, while the z-axis indicates depth.

### 3.1 Construction

**Sources** We leverage two datasets: (1) Visual Genome ([Krishna et al., 2017](#)), and (2) ADE20K ([Zhou et al., 2017](#)). The dense captions in Visual Genome provide a broad coverage of object classes, making it a suitable resource for collecting subtype candidates. For extracting hyponyms from knowledge base, we acquire "is-a" relations from ConceptNet ([Speer et al., 2017](#)), and augment the subtype candidate set. We extract spatial relations from ADE20K, as it provides images categorized by scene type – primarily indoor environments with high object density: *{bedroom, bathroom, kitchen, living room, office}*.

**Models** To collect subtype candidates (as detailed in §2.1), we perform part-of-speech tagging to extract object names (noun) from caption data, using LSTM-CRF ([Akbik et al., 2018](#)). Our subtype selection module is instantiated with UniCL ([Yang et al., 2022](#)) – designed for discriminative representations and broad semantic coverage of entities. To compute depth map from monocular image, we use DPT ([Ranftl et al., 2021](#)). To infer object color from image region, we query OFA ([Wang et al., 2022](#)), using the prompt template: "what color is the <object>?", and map zero-shot predictions to basic color set ([Berlin and Kay, 1991](#)) as detailed in Appendix [A.4](#).

### 3.2 Task Summary

Table [1](#) summarizes statistics for VIPHY, comprising the number of objects, classes and instances for each dimension. For multi-label tasks, we report the label cardinality, *i.e.* number of labels for a sample. We also indicate the number of objects with

| Subtype | objects | 14k |
| | objects with subtype | 1.8k |
| | subtype cardinality | 7.73 (12.91) |
| Color | objects (instances) | 14k |
| | classes | 11 |
| | label cardinality | 2.42 (1.37) |
| Size | objects | 1.6k |
| | instances | 43k |
| | relations | 2 |
| Spatial | objects | 300 |
| | instances | 6.5k |
| | relations | 3 |
| | label cardinality | 1.28 (0.45) |
| | scenes | 5 |

Table 1: Dataset statistics for VIPHY. The cardinality is reported with mean and standard deviation.

subtypes and subtype cardinality[6]. Note that while we extract size relations and color attributes from same source images (§3.1), we ignore *contextual* subtypes for size relations, as they serve a limited role towards informing object size (e.g. rain coat). However, as we only consider co-occurring objects, we implicitly incorporate context for objects in comparison, *i.e.* help disambiguate word sense. We collect 7.1k smaller and 14.5k larger relations, and balance labels by including their complements. The label distributions of color dataset is provided in Appendix Fig. 10.

**Formulation**    The objective of VIPHY tasks is to measure the ability to generalize physical knowledge pertaining to objects. To probe models with textual prompts, we map the raw distribution of labels (acquired by our pipeline) to typical values, as detailed in Appendix A.2. In the **color** task, objects can have multiple labels from the set of 11 basic colors as defined in Berlin and Kay (1991): *{red, orange, yellow, brown, green, blue, purple, pink, white, gray, black}*. Likewise, we consider multiple labels for **spatial** relations from {*below*, *above*, *similar level*}, conditioned on a scene type as mention in §3.1. Lastly, **size** relations are mapped to a single label from {*smaller*, *larger*}.

## 4   Experiments

We evaluate several state-of-the-art models under zero-shot and finetune settings (§4.1), and conduct further analysis of model performance (§4.2). The datasets are partitioned into 20% train, 10% dev,

70% test set.

**Baselines**    We consider the following language (LM) and vision-language (VLM) models:

- **LMs**: BERT (Devlin et al., 2019), RoBERTa (Liu et al., 2019), DeBERTa (He et al., 2020) and UnifiedQA (Khashabi et al., 2020).

- **VLMs**: VisualBERT (Li et al., 2019), ViLT (Kim et al., 2021), CLIP (Radford et al., 2021) and FLAVA (Singh et al., 2022).

**CapBERT**    In addition to the aforementioned baselines, we explore to what degree does an LM pretrained only on image captions, encodes visual knowledge. Such a model effectively serves as a diagnostic baseline – ablating the grounding mechanism in VLMs. We build CapBERT by pretraining BERT$_{base}$[7] on captions (4M) from COCO (Chen et al., 2015), CC3M (Sharma et al., 2018) and VG (Krishna et al., 2017) datasets. While a contemporaneous work by Zhang et al. (2022) developed a similar baseline, we find limited comparisons on color and no evaluation on their size dataset. Through extensive benchmarking, we are able to derive novel insights with CapBERT (§4.1).

**Finetuning**    To evaluate under finetune setting, we train a linear classifier on top of the model's output, while rest of the weights are frozen. We use Softmax Cross Entropy loss for single and multi-label setups, following Mahajan et al. (2018). All probes are finetuned for 50 epochs, with batch size of 8, using Adam optimizer (Kingma and Ba, 2015) and a learning rate of $10^{-4}$.

**Prompts**    For probing LMs and VLMs, we provide manually designed textual prompt as input to the model. The prompt templates for probing across color, size and spatial tasks, under zero-shot (ZS) and finetune (FT) settings are given in Table 2. Besides models trained on the masked language objective[8], the question-answering baseline (UnifiedQA) follows a common template[9] for both ZS and FT settings.

**Metrics**    We introduce the following metrics for measuring multi-label task performance:

---

[6]Note: The subtype cardinality is only reported for objects with subtype.

[7]Note that our VLM baselines follow the base model size. To ensure fair comparisons, we build the base version.

[8]CLIP being the exception, cannot be evaluated under ZS setting. Under FT, it uses `EOS` instead of `CLS` token.

[9]The prompt includes all classes as choices.

| Task | Setting | Prompt |
|------|---------|--------|
| Color | ZS | $O$ is of [MASK] color |
| | FT | [CLS] color of $O$ |
| | QA | What is the color of $O$? (a) .. (b) .. |
| Size | ZS | $O_1$ is [MASK] than $O_2$ in size |
| | FT | [CLS] size of $O_1$ in comparison to $O_2$ |
| | QA | what is the size of $O_1$ in comparison to $O_2$? (a) .. (b) .. |
| Spatial | ZS | in a $S$, the $O_1$ is located [MASK] the $O_2$ |
| | FT | [CLS] in a $S$, the $O_1$ is located in comparison to $O_2$ |
| | QA | in a $S$, where is $O_1$ is located in comparison to $O_2$? (a) .. (b) .. |

Table 2: Prompt templates across tasks and evaluation settings. Here, *O*, *R* and *S* are placeholders for object, relation and scene type respectively.

- Relaxed Accuracy (R-Acc) – The prediction ($P_i$) is accurate if the most probable label ($l_i$) belongs to the set of ground-truth labels ($T_i$).

$$RA = \sum_{i \in D} \frac{[l_i \cap T_i] \wedge [l_i = \arg\max_l P_i(l)]}{|D|}$$

- True Confidence (Conf) – The sum of predicted probabilities for labels in the ground-truth set.

$$C = \sum_{i \in D} \frac{\sum_{l \in T_i} P_i(l)}{|D|}$$

Here, $D$ denotes samples in the evaluation set. In addition to the aforementioned metrics, we also report the macro-averaged F1-score (F1).

**Human Performance** To provide an upper bound on VIPHY tasks, we use CoDa (Paik et al., 2021) for color – computed over 432 overlapping objects.[10] For size and spatial tasks, we evaluate 100 relations with three external annotators [11] (crowdsourced) and report the average scores.

## 4.1 Results

**Zero-Shot** We report zero-shot performance using R-Acc metric, across all tasks in Table 3. For spatial task, we only consider two labels from {*above, below*}, due to the limitation of single word masking in selected baselines. We observe significant variations in model performance across

tasks, with VLMs (VisualBERT) performing worse than their LM counterparts – underscoring the challenges of manual prompting (Jiang et al., 2020). The best scoring baseline (UnifiedQA) falls at least 30% points below human scores.

| Model | Color | Size | Spatial |
|-------|-------|------|---------|
| BERT$_{large}$ | 48.39 | 44.61 | 20.96 |
| RoBERTa$_{large}$ | 0.59 | 47.01 | 17.52 |
| UnifiedQA$_{large}$ | 51.00 | 51.76 | 63.04 |
| VisualBERT | 9.06 | 24.91 | 9.57 |
| Human | 97.45 | 90.12 | 88.24 |

Table 3: Zero-shot results (R-Acc) across all tasks.

**Finetune** When compared to zero-shot results, we report improved calibration under finetuned probing, as evident from results on color (Tab. 4), size (Tab. 6) and spatial tasks (Tab. 5). We find that VLMs score higher than LMs – specifically their "caption-only" counterpart (CapBERT) on the color task. These results hint at the role of color attribute in grounding entities. However, CapBERT outperforms VLMs on both size and spatial tasks, indicating that despite access to visual representations, VLMs do not retain such relational knowledge as effectively. In particular, it highlights that position encoded visual inputs in VLMs remain insufficient towards consolidating spatial knowledge. Lastly, CapBERT outperforming other LMs is likely due to the domain similarity between the pretraining source and the evaluation tasks[12].

## 4.2 Analysis

**Color: Cardinality** We further analyze model performance with respect to label cardinality (i.e. number of ground-truth colors for an object), by grouping objects accordingly. As shown in Fig. 5, we report results for three baselines, their average, along with human scores. While the performance is expected to increase with the cardinality[13], we notice an inconsistency between model and human scores. In particular, while difference in overall confidence scores (as inferred from Table 4) for human and the model average is ~18%, the relative differences between the two – ranges from ~12% (x = 6) to ~40% (x = 1), where x-axis denotes the label cardinality. While color influences object

---

[10]The label distributions of VIPHY and CoDa are provided in Appendix – Fig. 10 and Fig. 11

[11]The inter-annotator agreement measure (Fleiss' Kappa) for size and spatial are 0.85 and 0.78, respectively.

[12]For instance, a prepositional phrase can convey both abstract (*on schedule*) and physical (*on table*) relations, with captions predominantly containing the latter.

[13]R-Acc & Conf are 1, when cardinality is 11.

| Model | R-Acc | Conf | F1 |
|---|---|---|---|
| CapBERT | 70.45 | 58.55 | 40.91 |
| BERT$_{base}$ | 66.87 | 55.59 | 30.94 |
| RoBERTa$_{large}$ | 55.95 | 49.28 | 20.93 |
| UnifiedQA$_{large}$ | 62.34 | - | - |
| DeBERTa$_{xxl}$ | 72.74 | 59.59 | 36.33 |
| VisualBERT | 66.22 | 50.99 | 24.46 |
| ViLT | 64.83 | 53.92 | 30.27 |
| FLAVA | 76.33 | 62.84 | 38.74 |
| CLIP | **79.96** | 65.50 | 49.54 |
| Human $_{CoDa}$ | 97.45 | 78.65 | 68.82 |

Table 4: Color results.

| Model | R-Acc | Conf | F1 |
|---|---|---|---|
| CapBERT | **69.93** | 62.09 | 60.78 |
| BERT$_{base}$ | 67.25 | 59.91 | 61.34 |
| RoBERTa$_{large}$ | 54.88 | 58.40 | 58.88 |
| UnifiedQA$_{large}$ | 62.04 | - | - |
| DeBERTa$_{xxl}$ | 62.30 | 61.27 | 60.54 |
| VisualBERT | 63.08 | 58.40 | 58.88 |
| ViLT | 65.78 | 60.28 | 59.80 |
| FLAVA | 63.71 | 61.06 | 60.56 |
| CLIP | 65.10 | 63.56 | 62.26 |
| Human | 88.24 | - | 81.22 |

Table 5: Spatial results.

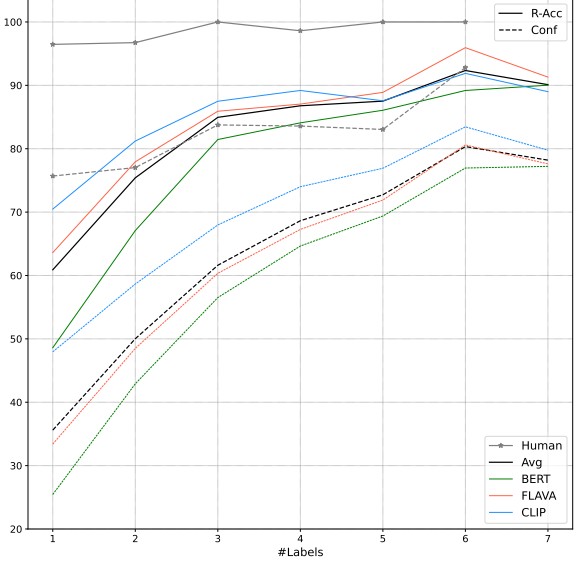

Figure 5: Effect of label cardinality (x-axis) on color prediction, as measured by R-Acc and Conf. The *Avg* curves (black) indicate average model performance.

perception in humans (Gegenfurtner and Rieger, 2000), these results show that VLMs do not ascribe a similar degree of saliency to color, especially for uni-color objects (*i.e.* cardinality of one).

**Color: Category-wise** To conduct error analysis on color task, we define categories by grouping ob-

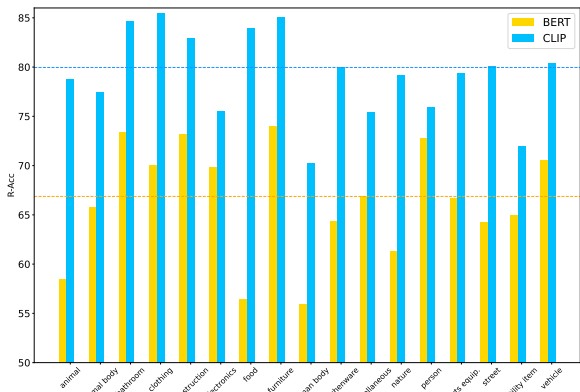

Figure 6: Category-wise performance on color (R-Acc). The dashed lines indicate average scores.

jects that belong to a common entity. We manually select category names from the object ontology provided in Open Images dataset (Kuznetsova et al., 2020). We assign each object to a category by computing the most similar category. We use Sentence-BERT (Reimers and Gurevych, 2019) and thus artificially convert each category to a sentence by concatenating the category name with few objects – serving as semantic cues (Appendix Tab. 8). We report category-wise performance (R-Acc) for BERT and CLIP, as provided in Fig. 6. We observe significant differences in performance between the two baselines, across categories. For *animal* and *nature* categories, BERT performs poorly with respect to the mean score. This difference widens for *food*, hinting at the effects of reporting bias as their typical colors are less likely to be written in text. The strong performance of CLIP for this category, however, indicates that such under reporting can be mitigated by visual modality. In contrast, for categories such as *electronics* and *vehicle*, we observe that LMs perform well – likely because color often plays a role in describing objects such as gadgets and cars.

**Size: Transitivity** As the size dataset[14] is composed of frequently co-occurring object pairs, we intend to evaluate model's ability to infer relative size for objects linked transitively across scenes. We build a new evaluation set comprising transitive relations from the standard size dataset, ensuring no overlapping instances between the two. The results (Tab. 6) indicate that LMs (on average) improve by significant margin, compared to VLMs. While the improvements on the evaluation set can

---
[14]For reference, the #instances for size evaluation sets are as follows – standard: 30k, subtype: 23k, transitive: 20k.

| Model | Standard | Subtype | Transitive |
|---|---|---|---|
| CapBERT | **83.69** | **79.14** | **91.82** |
| BERT$_{base}$ | 78.35 | 72.28 | 77.29 |
| RoBERTa$_{large}$ | 65.23 | 57.12 | 69.31 |
| UnifiedQA$_{large}$ | 62.20 | 60.66 | 90.78 |
| DeBERTa$_{xxl}$ | 74.73 | 66.88 | 69.79 |
| **LM$_{average}$** | 69.37 | 64.23 | 74.54 |
| VisualBERT | 76.99 | 64.00 | 77.69 |
| ViLT | 78.54 | 57.32 | 86.18 |
| FLAVA | 82.67 | 69.54 | 81.78 |
| CLIP | 75.43 | 66.56 | 72.48 |
| **VLM$_{average}$** | 79.15 | 64.35 | 79.53 |
| Human | 90.12 | - | - |

Table 6: Size results reported across different evaluation sets, measured by accuracy (random baseline: 50%).

.

be partially attributed to objects on the relatively extreme ends of size clusters being paired up, they are able to generalize on transitive relations.

**Size: Subtypes** While qualifiers denoting visual context tend to inform the typicality of color attribute, their effect on size is likely inconsequential. Therefore, models should retain their performance with reference to the standard set. We test this hypothesis by creating an evaluation set comprising contextual subtypes for objects in the standard test set. While the addition of subtypes leads to performance drop across all models (Tab. 6), we observe that LMs are more robust in comparison to VLMs.

## 5 Related Works

**Physical Commonsense** Recent years have witnessed a renewed interest in commonsense via natural language benchmarks (Talmor et al., 2019; Singh et al., 2021). Specific works have evaluated the language models on their ability to reason about physical commonsense (Bisk et al., 2020b; Qasemi et al., 2022), and identified reporting bias as a potential bottleneck (Forbes et al., 2019; Paik et al., 2021). In this work, we direct our focus towards vision-language models pretrained on large paired image-text datasets, and evaluate them on visually accessible commonsense knowledge. While prior works have probed knowledge pertaining to color (Paik et al., 2021; Mullenbach et al., 2019) and size (Talmor et al., 2020), their coverage of objects is severely limited in comparison to VIPHY (30×). Beyond size and color, Zhang et al. (2022) also incorporated object shape and material.

Recently, Liu et al. (2022a) have evaluated spa-

tial reasoning in images, spanning 65 relation types. In contrast, VIPHY measures the ability to recall spatial commonsense. Additionally, whereas Liu et al. (2022b) have probed spatial knowledge for human-object interaction (224 instances) under 15 action types (e.g. driving, cooking), we consider the spatial layout of objects across scene types over 6k instances, independent of events.

**Vision-Language Resources** While image classification (Deng et al., 2009) can be construed as one of the earliest attempts at bridging vision and language, recent years have witnessed a plethora of resources. Visual reasoning tasks have been directed towards object attributes (Antol et al., 2015), activities (Chen et al., 2015), as well as social (Zellers et al., 2019) and temporal commonsense (Fu et al., 2022). Recently, VLMs (Lu et al., 2019; Li et al., 2020; Radford et al., 2021) have demonstrated strong performance on such tasks. These works evaluate the requisite knowledge to reason about a specific instance, VIPHY in contrast probes the knowledge retained in the absence of visual context, *i.e.* generalized from instances.

**Knowledge in LMs** Recent advancements in language models (Devlin et al., 2019; Raffel et al., 2020), pretrained on large corpora, has led to significant improvements across several reasoning tasks (Wang et al., 2019). Prior works have also highlighted the capacity of these models to acquire several types of knowledge such as factual (Petroni et al., 2019; Roberts et al., 2020), instructional (Huang et al., 2022) and commonsense (Da et al., 2021). In this work, we study to what degree do their vision-language analogs (VLMs) – driven by the availability of massive paired image-text datasets, retain information that is easily accessible in images.

## 6 Conclusion

We present VIPHY, a large scale resource for probing "visible" physical knowledge – information easily accessible from images of static scenes, across dimensions of color, size and space. We design an automated pipeline to extract and consolidate such knowledge facts from images, and introduce a new resource for evaluating spatial knowledge of common environments. Our benchmarking evaluation highlights a huge gap between model and human performance across all three tasks. Furthermore, while prior works have reported VLMs

to be more effective, our caption pretrained baseline (CapBERT) significantly outperforms VLMs on the ability to recall size and spatial knowledge. These results underscore that despite access to visual modality, existing VLMs struggle to retain visual knowledge as effectively.

## Acknowledgement

We wish to acknowledge Bowen Zhang for his thoughtful comments on our paper. We also appreciate the constructive feedback from our anonymous reviewers, which has played a vital role in improving and clarifying this paper. This work is supported by the DARPA MCS program under Contract No.N660011924033 with the United States Office Of Naval Research.

## Ethical Implications

We build VIPHY from existing images from crowd-verified visual datasets which have been identified to lack geographical diversity, often limited to scenes from Europe and North America (Shankar et al., 2017). Furthermore, such datasets are subjected to several kinds of biases at different stages of collection and annotation such as selection bias, framing bias and observer bias (Fabbrizzi et al., 2022). Therefore, its likely that such biases will be reflected in our dataset as well. As we also report benchmarking results on VIPHY, the model performance may not be reflected as accurately on knowledge pertaining to different geographical and cultural backgrounds, as studied by Yin et al. (2021). Lastly, our proposed resource is limited to English, and thus excludes any considerations for multilingual models (Yin et al., 2022).

## Limitations

We extract spatial knowledge directly from images, we assume that the camera's image plane is somewhat orthogonal to the ground (transverse plane), and do not account for the edge cases where the image plane will be parallel, *i.e.* top view. For collecting subtype candidates, we parse Visual Genome (Krishna et al., 2017) captions to acquire object names. To assign object subtypes, we rely on pretrained vision-language model (UniCL; Yang et al., 2022) which can be a likely source of noise in the pipeline. Furthermore, quantifying their performance on subtype selection task is beyond the scope of our work, due to unavailability of ground-truth annotations. In contrast to previous works,

since we automatically estimate object size from the image – the estimations are limited by the quality of bounding boxes and depth predictions from models.

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

# Appendix

## A Implementation Details

### A.1 Pipeline Parameters

To cluster objects for computing relative size, we use Jenks Natural Breaks (Jenks, 1967), with #clusters = 5 following the manual groupings of object sizes in Liu et al. (2022b). We also experimented with #clusters = 3, but qualitatively observed less optimal clusters. In spatial module, we create 3 partitions of uniform size and overlap.

### A.2 Typical Labels

We derive typical labels from the raw label probabilities ($C$) by filtering classes as per a predefined threshold $p_{min}$, that can be interpreted as either noise or rare occurrence. Formally, we apply the filter as follows: $C = \{(c, p) | p > p_{min}, (c, p) \in C\}$. Here, $p_{min}$ is defined as:

$$
\begin{cases}
10\% & 4 \leq |C| \leq 11 \\
20\% & |C| = 3 \\
30\% & |C| = 2
\end{cases}
$$

The resulting distribution is re-normalized and the filtering step is applied recursively.

### A.3 Defining Spatial Relations

Our objective is to map the raw coordinates in an image for two objects to discrete relations. We define a simple set of rules to convey *above* and *similar* level, from object annotations. Given bounding box or polygon mask, we first compute its centroid along with the lowest point. We then compare the y-coordinates of objects as illustrated in Fig. 7. If an object's lowest point is above the other's centroid, we map it to *above*, else *similar* level.

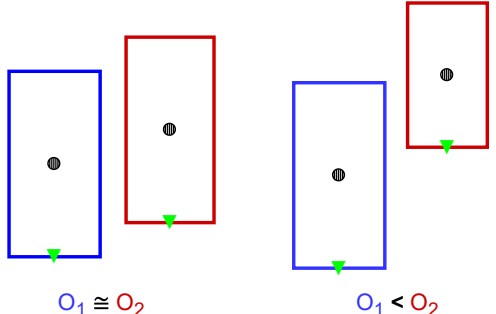

Figure 7: Illustrates our definition of spatial relation from raw annotations.

### A.4 Prediction to Basic Color

We use OFA in our pipeline, and map generated text to the basic color set as shown in Table 7. If the model predicts multiple colors, we assign each of them to the object instance.

| Basic Color | Raw Predicted Terms |
|---|---|
| Yellow | *gold, golden, blonde, beige, peach, cream* |
| Brown | *wooden, tan, beige, bronze, copper* |
| Gray | *grey, silver, metal, steel* |
| Pink | *peach* |
| Purple | *violet* |
| Red | *maroon* |
| Green | *teal* |
| Blue | *teal, turquoise* |

Table 7: Mapping between raw predictions (OFA) and basic color terms.

### A.5 Few-Shot Performance

We report few-shot results for color (Fig. 8) and size (Fig. 9) task, by building variants of the train set with respect to the percentage of samples. The improvement in VLMs and CapBERT likely indicates prompt adaptation due to their relatively weaker linguistic ability – compared to LMs.

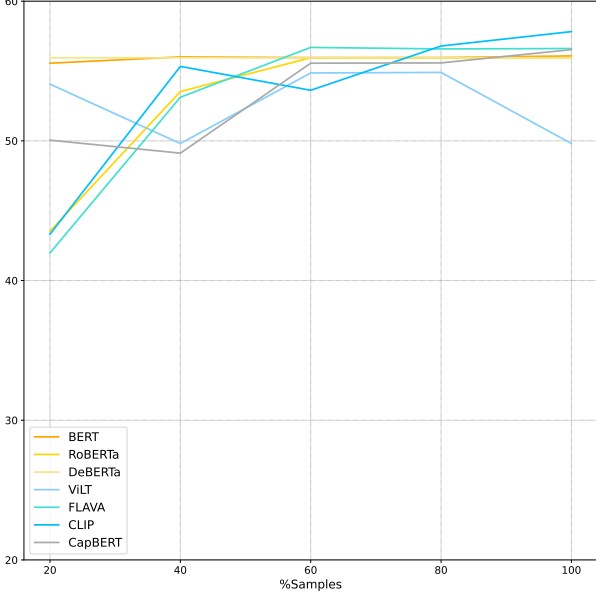

Figure 8: Few-shot performance on color with respect to percentage of samples in the train set.

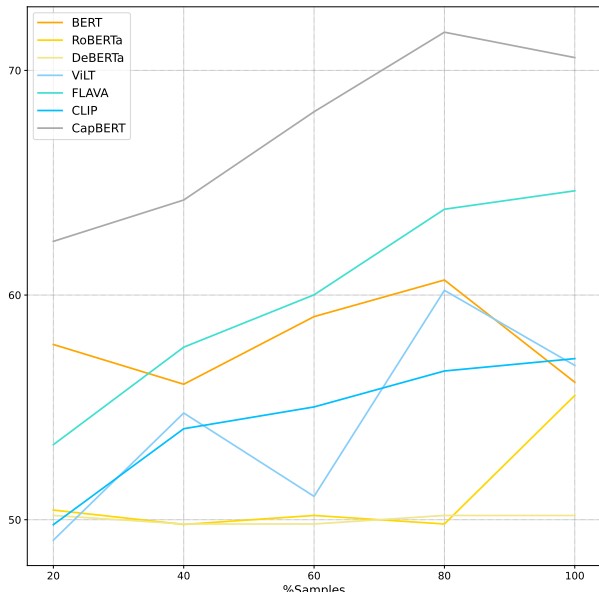

Figure 9: Few-shot performance on size.

| Category | Objects |
|---|---|
| person | man, woman, lady, boy, girl, child |
| human body | hand, mouth, hair, teeth, skin |
| animal | bird, fish, reptile, insect, cattle, pet |
| animal body | paw, hooves, tail, mane, fur, whisker |
| nature | sky, tree, flower, sea, beach, snow |
| food | vegetable, fruit, dessert, snack |
| furniture | table, chair, shelf, bed, cabinet |
| kitchenware | oven, fridge, knife, bowl, sink |
| bathroom | towel, shower, toothbrush, toilet |
| sports | helmet, racket, ball, glove, skateboard |
| clothing | hat, shirt, pant, skirt, shoe, glasses |
| electronics | laptop, mouse, printer, cell, projector |
| vehicle | bike, car, bus, truck, boat, plane |
| street | traffic, sign, crosswalk, hydrant, pole |
| construction | store, building, school, airport, bridge |
| utility | lamp, paper, trashcan, bag, extinguisher |
| miscellaneous | entity, object, thing, stuff, item |

Table 8: Categories and objects used as semantic cues. We map them to sentence with "<Category>: <Objects>" template.

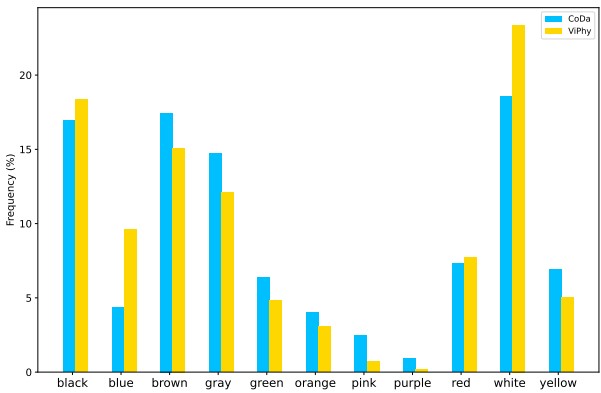

Figure 10: Color distribution of objects in VIPHY and CoDa.

| Context | Examples |
|---|---|
| Event | *wedding cake, bathing soap* |
| Location | *kitchen sink, street lamp* |
| State | *bare tree, sliced apple* |
| Part | *piano key, bike wheel* |

Table 9: Context-based subtype examples

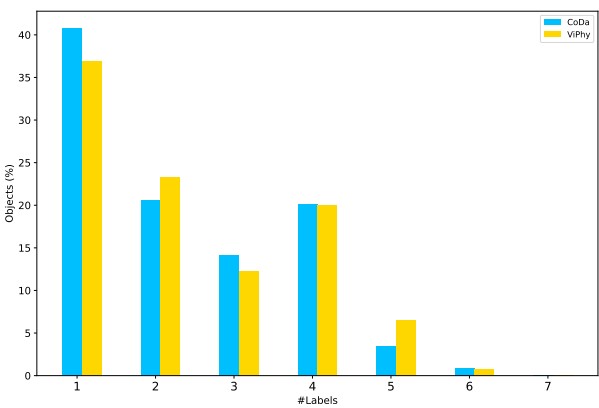

Figure 11: Color distribution with respect to label cardinality in VIPHY and CoDa.

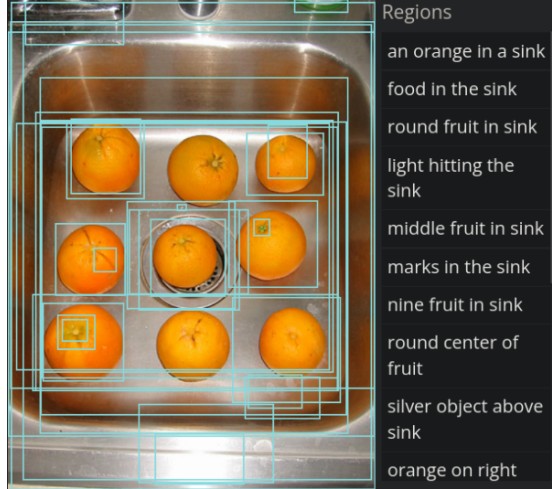

Figure 12: A sample image and corresponding captions from Visual Genome dataset. Illustrates how humans omit the subtype *kitchen* sink, when annotating images.

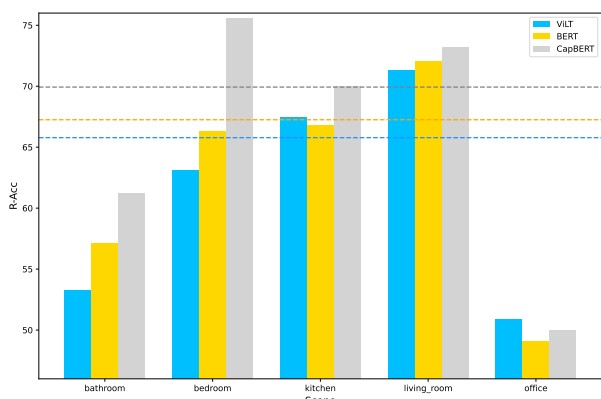

Figure 13: Scene-wise performance on spatial task, with dashed lines indicating the overall scores. The sample distribution is 12%, 7%, 42%, 34%, 2%.

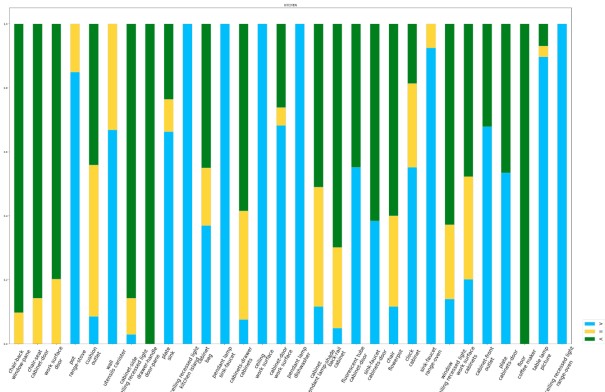

Figure 14: Spatial relation distribution for object pairs in *kitchen* scene from VIPHY.

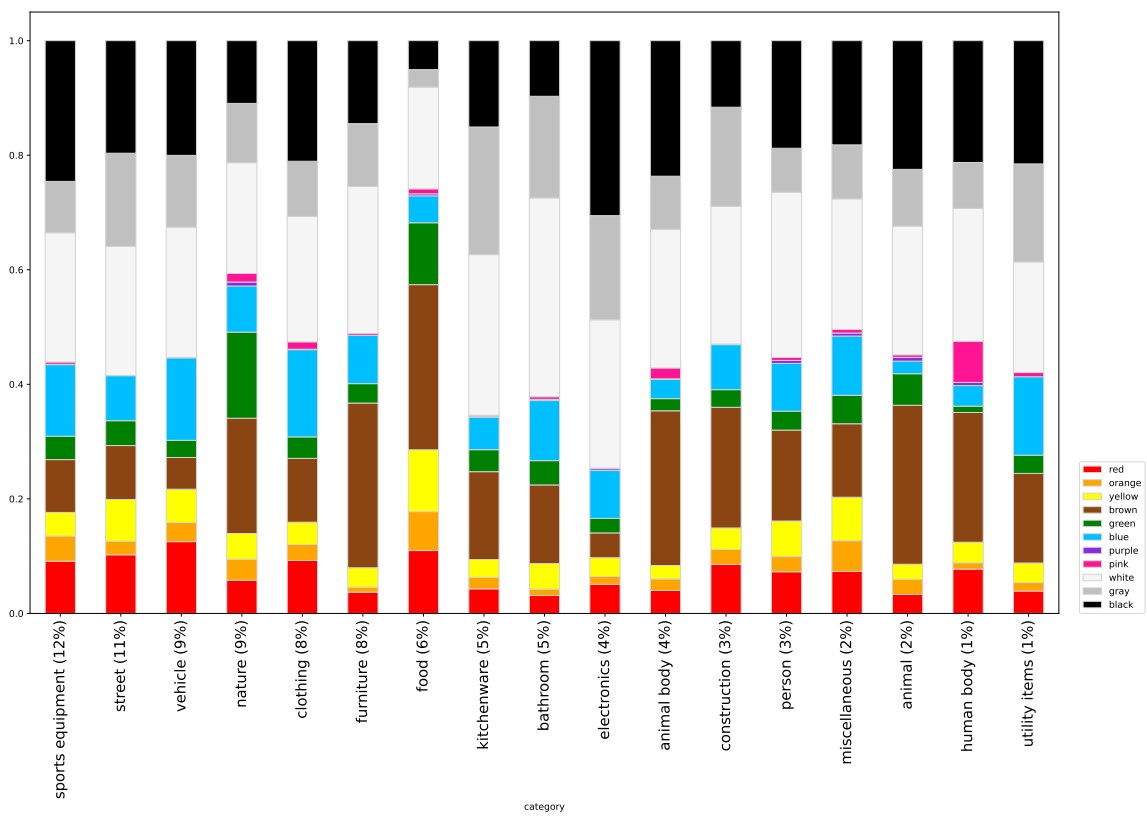

Figure 15: Color distribution for object categories as defined in §4.2. The parenthesized values indicate the percentage of samples in each category.

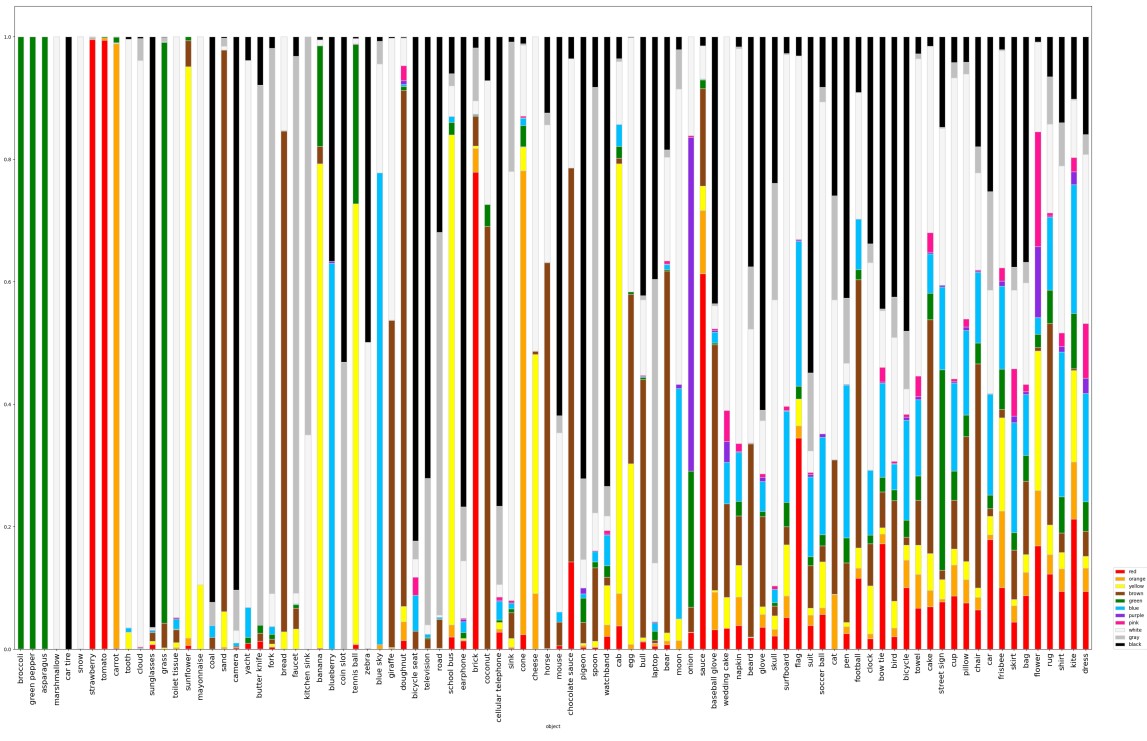

Figure 16: Color distribution for 90 objects from VIPHY, sorted by entropy.