# OpenReview forum: "VIPHY: Probing “Visible” Physical Commonsense Knowledge"
_EMNLP/2023/Conference — EMNLP 2023 Findings_

### Official Review · Reviewer_Vefi · 2023-08-02

**Soundness:** 3

**Excitement:**

3: Ambivalent: It has merits (e.g., it reports state-of-the-art results, the idea is nice), but there are key weaknesses (e.g., it describes incremental work), and it can significantly benefit from another round of revision. However, I won't object to accepting it if my co-reviewers champion it.

**Paper Topic And Main Contributions:**

This work explores the problem of physical knowledge acquisition of VLMs.
Main contributions:
1. Proposed an automatic pipeline for physical knowledge data generation
2. Constructed VIPHY dataset for physical knowledge based on existing Genome and ADE20K dataset
3. Evaluated a list of LMs and VLMs and reported performance comparison


**Reasons To Accept:**

1. The problem about physical knowledge acquisition and reasoning of VLM is generally interesting.
2. The work contributed a new dataset in this line of work.


**Reasons To Reject:**

Latest VLMs such as InstructBLIP, LLaVA not included for experiment which makes the claim that existing VLMs struggle to effectively consolidate physical knowledge less appealing.

**Reproducibility:**

4: Could mostly reproduce the results, but there may be some variation because of sample variance or minor variations in their interpretation of the protocol or method.

**Reviewer Confidence:**

3: Pretty sure, but there's a chance I missed something. Although I have a good feel for this area in general, I did not carefully check the paper's details, e.g., the math, experimental design, or novelty.

---

> ### Author Rebuttal · Authors · 2023-08-29
>
> We sincerely appreciate your review. Please find responses to your points below.
>
> ### Latest VLMS
> We agree with the reviewer that the evaluation could be more comprehensive by including more recent VLMs. However, the goal of our study is to evaluate the effectiveness of access to visual modality in VLMs in the context of visible commonsense reasoning. We show that the model trained on captions (CapBERT) outperforms the VLMs that we evaluated. This suggests that access to visual modality is not always necessary for good performance on visible commonsense reasoning tasks.
> We believe that the four VLMs that we evaluated represent a diverse range of architectures and capabilities and that our results are still valid even if we do not include the most recent models.
> We would be happy to include more recent VLMs in our evaluation in future work. However, we believe that our current results are still valuable and contribute to the understanding of the role of visual modality in VLMs for visible commonsense reasoning

---

### Official Review · Reviewer_2fNQ · 2023-08-10

**Soundness:** 3

**Excitement:**

4: Strong: This paper deepens the understanding of some phenomenon or lowers the barriers to an existing research direction.

**Justification For Ethical Concerns:**

No ethical issues.

**Missing References:**

N/A

**Paper Topic And Main Contributions:**

This paper studies an interesting field, how to preceive the physical commonsense knowledge from images via vision-language model.
From the technique side, the proposed framework first need to estimate both depth information and fine-grained categories from the input image.
Then, the depth and object size information is used for clustering, and the color information is acuqired from VLM.
The authors leverage ADE20K and Visual Genome for the experiments, which are evaluated on multiple state-of-the-art methods on both zero-shot and fine-tuning settings.

**Questions For The Authors:**

I appreciate this work, and I really hope the authors can address my concerns in the rebuttal, especially:

**Q1**: More implementation details of the proposed framework. For example, what object detectors do the authors use for cluster relation? What depth estimation method does the authors use?

**Q2**: More evidence or justification, to bridge the gap between 'Physical Commonsense Knowledge' and more rigid definition from image formulation and preception.

**Q3**: Some other presentation issues, listed in the presentation section.

**Reasons To Accept:**

- This work proposes a new task to estimate the attributes such as size, color and space from images, by leveraging VLM and other vision tools.

- The experiments are very extensive and solid.

- This paper is well-written and easy-to-follow.

- The authors provide both source code and data along with the submission, which I appreciate a lot. I hope it can be made publicly avaible, once published.

**Reasons To Reject:**

- Some implementation details inside the proposed framework are not very informative from the submission. For example, what object detectors do the authors use for cluster relation? What depth estimation method does the authors use?

-  As I am not an expert in NLP but in Computer Vision, especially with some low-level vision background, the claim of 'Physical Commonsense Knowledge' does not seem very comfortable for me. In fact, from the image preception and image formulation prespective, the physicial attributes may actually contain more attributes than color, space and size.
So, it would be highly appreciated if the authors can enhance this part, and provide some references to justify that, these three key factors can also align with the definition of physcial knowledge in vision community.


- Some other minor issues on presentation, please refer to the 'presentation improvement' section.

**Reproducibility:**

4: Could mostly reproduce the results, but there may be some variation because of sample variance or minor variations in their interpretation of the protocol or method.

**Reviewer Confidence:**

4: Quite sure. I tried to check the important points carefully. It's unlikely, though conceivable, that I missed something that should affect my ratings.

**Typos Grammar Style And Presentation Improvements:**

There are indeed some presentation issues.

- First, and the most important one. Page1, below the title. Why 'Anonymous ACL submission'?

- Each equation does not assign its number.

- From the reviewer's background, it would be better to present each function operation by using \rm function in latex.

- Fig.4 is not very informative. What does each color mean is not reflected in either legend or caption.

---

> ### Author Rebuttal · Authors · 2023-08-29
>
> We sincerely appreciate your review and your detailed comments for improving this work. Please find responses to your points below.
> ### Implementation Details
> We respectfully want to point out L257 “To compute depth map from monocular image, we use DPT (Ranftl et al., 2021)” which addressed your point on depth estimation. Regarding the object detector, we relied on the bounding boxes provided by the visual genome L238.
>
>
> ### Related studies from CV
> We appreciate your feedback on the parallel terms from the computer vision community. In this work, we have focused on a simple subset of visible commonsense that can be easily inferred from static images. However, we acknowledge that the common sense associated with the physical attributes of objects is more extensive than the three categories we have considered. We will update the manuscript to include more relevant literature on "physical knowledge."

---

### Official Review · Reviewer_jjqR · 2023-08-11

**Soundness:** 3

**Excitement:**

2: Mediocre: This paper makes marginal contributions (vs non-contemporaneous work), so I would rather not see it in the conference.

**Paper Topic And Main Contributions:**

VIPHY is a dataset measuring vision language models' understanding of visible physics commonsense knowledge, specially on color, size, and spatial relations of objects in the image. The dataset collection pipeline is done automatically without human annotation which first involves extracting object names and subtypes from existing captioning dataset, such as Visual Genome and ADE20K. Then, VLMs are used to infer colors object regions, size and spatial relations are inferred based on the ratio of depth estimation and bounding box area. Experiments probing the vision language model in their benchmark show a huge gap between model and human performance across all the tasks. In addition to vision-language model, the authors explore training a language only model on captioning dataset which achieve superior results in inferring size and spatial knowledge than VL models.

**Questions For The Authors:**

A. How do you validate that the instances are "correct"? For example, for size questions, suppose you see a zoomed-in cup in front of you and part of a car visible in the background. Running the label construction pipeline that relies on depth estimation would still consider the cup as "larger" than the "car". This is most likely the reason for relatively low oracle score in human performance in Table 5, where certain instances simply contain labels that are not true based on the human level of commonsense understanding.

B. What is the human acceptability / validation rate of collected instances?

C. How does this work provide more value than ViComte which follow similar automatic pipeline and same data source (Visual Genome)?

D. What is the human performance when they are not looking at the image?

E. What is the performance of current SoTA LLMs such as ChatGPT?





**Reasons To Accept:**

 - Introduces an innovative pipeline to automatically construct visual physics commonsense dataset that involves use of diverse tools: parser, depth estimation, vision language models, bounding box areas.
- Presents empirical findings that the vision language models show far worse performance in VIPHY than humans and calls for the need to address visible physics common understanding of such models to the NLP community.
- Demonstrates an in-depth evaluation of the performance of current vision language models, taking into account various factors such as types of questions, color categories, and label cardinality.

**Reasons To Reject:**

- No human validation of automated process which might be prone to error due to the following reasons:
  - Colors are derived from prediction of VL model, specifically OFA that might not yield a correct answer.
  -  Size relies on depth estimation models and bounding box areas to get the label. How do you account for the  case when smaller objects are zoomed in for size questions (See the example in Question A)?
  - Spatial: how do you validate that the objects are similar level with one another vs "smaller/larger". What if objects are in front of / behind one another, and have different levels of bounding box coordinates?

- Unclear motivation of dataset collection for visible physics commonsense v.s. prior work:
  - Dataset is constructed based on VisualGenome, which already include color and spatial information (above, under, behind, in front of, etc) in their scene graphs attributes and relation labels, and size can be trivially inferred from object labels most of the times. Why can we not just evaluate on a subset of VisualGenome scene graph tasks that deal with colors and spatial information?
  - Winoground [1] was carefully hand-curated by expert annotators instead of relying on model-based predictions, and include far rich annotations testing visual-language compositionality that actually require looking at images to do well.
  - ViComte [2] already covers similar and diverse visual commonsense knowledge, such as color, shape, material, size, and visual co-occurrence in their dataset. How does the authors' work provide more benefits over this dataset?

- Limited coverage of "visible commonsense" understanding. Color seems to be purely vision task, and size and spatial most of the times can be inferred from text information. There are far more interesting cases that test such knowledge, such as counterfactual reasoning asking (what would happen if this action is applied to the object in this image).
- The findings of CapBERT is not novel as it is already explored in Zhang et. al [2]. and the paper simply runs the same set up to validate their dataset.
  - This in fact further highlights that this dataset is essentially prone to **severe text biases, in which visual understanding is not required to perform well for size and spatial questions**. It is difficult to claim that the dataset tests *visible* physics commonsense knowledge if such information is not needed.
- Language model experiments are not as comprehensive and do not not cover performance of stronger LLMs such as ChatGPT. It would be more interesting for the language community to understand in their physics commonsense understanding skills.
- Experiment results illustrate that the task does not have much too improve on from using current SoTA models. Color questions are extracted from OFA models, and size + spatial questions do not require visual understanding to perform well. Couldn't then the task be solved by 1) run OFA (which is not evaluated on this dataset) for color questions, 2) run LLMs such as GPT-4 for size + spatial reasoning tasks.
- More qualitative examples of when images are important to do well in the task.

[1]: Winoground: Probing Vision and Language Models for Visio-Linguistic Compositionality [Thrush et. al.]

[2]: Visual Commonsense in Pretrained Unimodal and Multimodal Models [Zhang et. al.]

**Reproducibility:**

4: Could mostly reproduce the results, but there may be some variation because of sample variance or minor variations in their interpretation of the protocol or method.

**Reviewer Confidence:**

3: Pretty sure, but there's a chance I missed something. Although I have a good feel for this area in general, I did not carefully check the paper's details, e.g., the math, experimental design, or novelty.

---

> ### Author Rebuttal · Authors · 2023-08-29
>
> We sincerely appreciate your review. Please find responses to your points below.
> ### No Human Validation
> We agree with your point that any automated pipeline is susceptible to error. However, as shown in the human evaluations of Table 3, the data generated by ViPHY is of high quality. We have taken steps to mitigate the effects of noise on the data, such as feeding the bounding box of each object to the OFA model, which helps to improve the accuracy of color extraction. Additionally, the evaluation focuses on transitivity through pair-wise comparison, which minimizes the impact of size errors. Finally, the depth grouping algorithm ensures that objects are only compared in relatively equal depth, which eliminates cases where relative proximity results in an inflated size measurement.
> We did not feel it would be interesting to the general community to include a detailed ablation study and opted for an overall human evaluation of the data.
>
>
> ### Unclear Motivation
> The main strength of our first contribution, the ViPHy pipeline, is its scalability to new objects and sizes in all three tasks (30 times more objects than any existing resource) while preserving a reasonable level of quality for visible commonsense reasoning (human evaluation in Table 3). The only limitation of ViPHy's automatic pipeline is its reliance on bounding boxes (Figure 3).
> Although there are available resources on visible commonsense knowledge, such as Visual Genome and Winoground, they are all created using direct human annotations, which makes them both expensive to create and unscalable to new objects. In contrast, ViPHy's automatic pipeline is significantly more scalable and cost-effective.
> We respectfully disagree with your formulation that our "dataset is constructed based on VisualGenome". We solely use the images from VisualGenome due to their high-quality of bounding boxes. However, ViPHy is not limited to using VisualGenome images. In principle, ViPHy can be used to reproduce any resource, such as VisualGenome, without the need for costly human annotations.
>
>
> ### Limited Coverage
> We agree with your suggestion of using other setups, such as counterfactuals, for evaluating "visible commonsense" understanding. However, we believe that our approach of focusing on the simplest aspects of visible commonsense is a valid one, as it allows us to identify the shortcomings of SoTA VLMs in reasoning with these basic concepts.
> We also respectfully disagree with your formulation that color, size, and spatial knowledge are not part of commonsense knowledge. In fact, numerous prior works have classified these concepts as part of commonsense knowledge (Ilievski et. al. "Dimensions of commonsense knowledge." Knowledge-Based Systems 229 (2021)). Therefore, we believe that our work provides a valuable contribution to the field of commonsense reasoning by demonstrating the importance of these seemingly simple concepts.
>
>
> ### CapBERT and Text-bias in data
>
> We respectfully point out a misunderstanding of our work. All of the analyses in Section 4.1 are conducted in a text-only setup. In this work, we first use an automatic pipeline to extract visible commonsense knowledge from images. We then use the extracted knowledge (not the images) to probe two groups of models: language models (LMs) that are only trained with text modality, and visual language models (VLMs) that observe the world during their training through both visual and textual modalities. The goal of this experiment is to evaluate and compare VLMs' understanding of visible commonsense knowledge. We will make sure to clarify this in the final version.
> Therefore, we respectfully disagree with your argument that there is a text-only bias in our work and your statement that "visual understanding is not required".
> We also note that you have mentioned that models like CapBERT have been previously used in similar setups. However, we believe that our work is novel as our evaluation is on a larger scale and on the three dimensions: color, size, and space.
>
>
> ### LLMs
> We agree with your point on the need for the addition of LLMs such as ChatGPT as a baseline. Unfortunately, we were not able to finalize such results during the rebuttal period but they will be added to the final manuscript.

---

### Official Review · Reviewer_btZN · 2023-08-12

**Soundness:** 4

**Excitement:**

2: Mediocre: This paper makes marginal contributions (vs non-contemporaneous work), so I would rather not see it in the conference.

**Missing References:**

[1] Lu, Jiasen, et al. "Unified-io: A unified model for vision, language, and multi-modal tasks." arXiv preprint arXiv:2206.08916 (2022).

**Paper Topic And Main Contributions:**

This paper proposes a large, machine-generated dataset for evaluating Vision-language Models (VLMs) performance related to object color, size and spatial relationships. The result shows limitations of existing VLMs especially in spatial relation reasoning tasks.

**Questions For The Authors:**

Question A: In the evaluation method related to unmasking a token, is there any method to deal with synonymous tokens and prevent them from polluting the evaluation score?

Question B: Does the analysis and conclusion still hold for recent models like OFA? And what about even more recent models such as LLaVA (which is within 3 months and is totally okay to ignore this part)?

Question C: What are the strengths of this dataset compared to previous smaller, manually collected datasets? Can you illustrate these strengths with examples?

**Reasons To Accept:**

A. The proposed dataset is relatively large in size and  might be a helpful resource to the community.
B. The writing is clear.

**Reasons To Reject:**

A. The evaluation lacks inclusion of recent models, which may limit the comprehensiveness of the assessment. Among the four VLMs evaluated, only one is from 2022, while the rest are from earlier years. Including more recent, potentially stronger models like OFA, Unified-IO etc., would make the analysis and conclusions more robust and up to date.

B. Despite the author's emphasis on the dataset's size as a novelty, previous benchmarks have derived similar conclusions and analyses using smaller, manually curated datasets as partially noted in the Related Works section. The paper does not clarify how utilizing a larger, potentially noisier dataset adds unique value or insights to the community.

**Reproducibility:**

5: Could easily reproduce the results.

**Reviewer Confidence:**

3: Pretty sure, but there's a chance I missed something. Although I have a good feel for this area in general, I did not carefully check the paper's details, e.g., the math, experimental design, or novelty.

**Typos Grammar Style And Presentation Improvements:**

It would be more illustrative to include some sampled responses from the models in the body or appendix of the paper.

---

> ### Author Rebuttal · Authors · 2023-08-29
>
> We sincerely appreciate your review and detailed comments on our work. Regarding your presentation suggestions, we will add more examples from the data to the appendix. Please find responses to your other points below.
>
> ### Additional VLMs
> We agree with the reviewer that the evaluation could be more comprehensive by including more recent VLMs. However, the goal of our study is to evaluate the effectiveness of access to visual modality in VLMs in the context of visible commonsense reasoning. We show that the model trained on captions (CapBERT) outperforms the VLMs that we evaluated. This suggests that access to visual modality is not always necessary for good performance on visible commonsense reasoning tasks.
> We believe that the four VLMs that we evaluated represent a diverse range of architectures and capabilities and that our results are still valid even if we do not include the most recent models.
> We would be happy to include more recent VLMs in our evaluation in future work. However, we believe that our current results are still valuable and contribute to the understanding of the role of visual modality in VLMs for visible commonsense reasoning.
>
> ### Contribution and Strength of Resource
> Our first contribution is the automated pipeline that bypasses crowdsourced annotations to cover 30 times more objects than any existing resource. The only limitation of our pipeline is the existence of a high-quality bounding box and object detection that given current SoTa is cheap to acquire, hence ViPHy can scale up to new objects without the need for expensive human supervision.

---

### Meta-Review · Area_Chair_ZKoC · 2023-09-17

**Recommendation:** 3

**Metareview:**

**Summary:**
This paper proposes a large, machine-generated dataset, called VIPHY, aimed at evaluating Vision-language Models (VLMs) performance related to object color, size and spatial relationships. The result shows limitations of existing VLMs especially in spatial relation reasoning tasks. The automatic pipeline extracts object names and subtypes from existing captioning dataset, such as Visual Genome and ADE20K.
The authors evaluate the results on multiple state-of-the-art methods on both zero-shot and fine-tuning settings. Experiments probing the vision language model in their benchmark show a huge gap between model and human performance across all the tasks. In addition to vision-language model, the authors explore training a language only model on captioning dataset which achieve superior results in inferring size and spatial knowledge than VL models.

**Strengths:**
The reviewers agree on the following strengths:
1. The authors have introduced a substantial dataset that holds the potential to become a valuable resource for the community.
2. Their work presents an innovative pipeline for the automatic construction of a visual physics commonsense dataset, leveraging diverse tools such as a parser, depth estimation, vision language models, and bounding box areas. Building upon these foundations, the contribution introduces a novel task: estimating attributes like size, color, and space from images.
3. The authors provide empirical evidence demonstrating that vision language models perform considerably worse in VIPHY compared to humans. This underscores the imperative need to address the common understanding of visible physics in such models within the NLP community.
4. Lastly, the authors have thoughtfully included both source code and data with their submission. It would be worthy if these resources could be made publicly available upon the paper's publication.

**Weaknesses:**
Reviewers have identified the following weaknesses in the paper:
1. The work lacks the inclusion of recent models, potentially limiting the comprehensiveness of the assessment. The absence of stronger models like OFA and Unified-IO harms the robustness of the analysis. Moreover, this absence weakens the claim that existing Visual Language Models (VLMs) struggle to effectively integrate physical knowledge. Actually, it would be more intriguing for the language community to gain insights into the capabilities of Large Language Models (LLMs) in the domain of physics commonsense understanding.
2. The paper does not clearly articulate how utilizing a larger, “potentially noisier” dataset adds insights to the community. In fact, previous benchmarks have reached similar conclusions and analyses using smaller, manually curated datasets.
3. Reviewers have emphasized the absence of human validation in the automated process, which may introduce errors for several reasons. Additionally, the paper has limited coverage of "visible commonsense" understanding.
4. The existence of CapBERT highlights that the dataset may be prone to significant text biases. This means that visual understanding may not be necessary for size and spatial questions, leading doubt on the claim that the dataset effectively tests visible physics commonsense knowledge.

**Author-Reviewer discussion and acknowledgment:**
Reviewers raised various questions and concerns to which the authors responded during the rebuttal and discussion phase, outlining planned improvements and engaging in further discussions with the reviewers. All reviewers have responded and acknowledged the authors' arguments, while still pointing out some weaknesses in the paper.

**Conclusion:**
The writing is clear, and the paper is well-written and easy to follow. However, reviewers suggest that the authors address the identified typos. Furthermore, reviewers recommend that the authors include additional references and improve the paper based on the points raised during the discussion phase.

---

### Decision · Program_Chairs · 2023-10-07

**Decision:**

Accept-Findings

**Comment:**

**Summary:**
This paper proposes a large, machine-generated dataset, called VIPHY, aimed at evaluating Vision-language Models (VLMs) performance related to object color, size and spatial relationships. The result shows limitations of existing VLMs especially in spatial relation reasoning tasks. The automatic pipeline extracts object names and subtypes from existing captioning dataset, such as Visual Genome and ADE20K.
The authors evaluate the results on multiple state-of-the-art methods on both zero-shot and fine-tuning settings. Experiments probing the vision language model in their benchmark show a huge gap between model and human performance across all the tasks. In addition to vision-language model, the authors explore training a language only model on captioning dataset which achieve superior results in inferring size and spatial knowledge than VL models.

**Strengths:**
The reviewers agree on the following strengths:
1. The authors have introduced a substantial dataset that holds the potential to become a valuable resource for the community.
2. Their work presents an innovative pipeline for the automatic construction of a visual physics commonsense dataset, leveraging diverse tools such as a parser, depth estimation, vision language models, and bounding box areas. Building upon these foundations, the contribution introduces a novel task: estimating attributes like size, color, and space from images.
3. The authors provide empirical evidence demonstrating that vision language models perform considerably worse in VIPHY compared to humans. This underscores the imperative need to address the common understanding of visible physics in such models within the NLP community.
4. Lastly, the authors have thoughtfully included both source code and data with their submission. It would be worthy if these resources could be made publicly available upon the paper's publication.

**Weaknesses:**
Reviewers have identified the following weaknesses in the paper:
1. The work lacks the inclusion of recent models, potentially limiting the comprehensiveness of the assessment. The absence of stronger models like OFA and Unified-IO harms the robustness of the analysis. Moreover, this absence weakens the claim that existing Visual Language Models (VLMs) struggle to effectively integrate physical knowledge. Actually, it would be more intriguing for the language community to gain insights into the capabilities of Large Language Models (LLMs) in the domain of physics commonsense understanding.
2. The paper does not clearly articulate how utilizing a larger, “potentially noisier” dataset adds insights to the community. In fact, previous benchmarks have reached similar conclusions and analyses using smaller, manually curated datasets.
3. Reviewers have emphasized the absence of human validation in the automated process, which may introduce errors for several reasons. Additionally, the paper has limited coverage of "visible commonsense" understanding.
4. The existence of CapBERT highlights that the dataset may be prone to significant text biases. This means that visual understanding may not be necessary for size and spatial questions, leading doubt on the claim that the dataset effectively tests visible physics commonsense knowledge.

**Author-Reviewer discussion and acknowledgment:**
Reviewers raised various questions and concerns to which the authors responded during the rebuttal and discussion phase, outlining planned improvements and engaging in further discussions with the reviewers. All reviewers have responded and acknowledged the authors' arguments, while still pointing out some weaknesses in the paper.

**Conclusion:**
The writing is clear, and the paper is well-written and easy to follow. However, reviewers suggest that the authors address the identified typos. Furthermore, reviewers recommend that the authors include additional references and improve the paper based on the points raised during the discussion phase.